# Automated Stateful Specialization for Adaptive Agent Systems

**Myan Vu**[1]*    **Harrish Ayyanar**[2]    **Jiang Pang**[3]    **Anwiketh Reddy**[4]    **Mayank Goel**[5]

[1]University of Auckland    [2]University College London    [3]Algoverse    [4]Northeastern University    [5]Lossfunk

## Abstract

Current automated agent design frameworks produce either static workflows that lack adaptability or per-query optimizers that prevent the accumulation of deep, agent-level task expertise. We propose a new direction that reconciles these paradigms: creating stateful teams of specialist agents that accumulate knowledge over time and can be reconfigured for novel tasks entirely without human intervention. To this end, we introduce ASPEC, a framework that manages this full agent lifecycle by first autonomously **discovering** specialist archetypes via evolutionary search and then **cultivating** their expertise through experience, mirroring how human experts learn through practice and reflection. We further introduce a lightweight hierarchical control policy, "retain-then-escalate," which governs when to leverage the established agent system versus when to adapt its structure. Through comprehensive experiments, we demonstrate that this approach leads to significant performance gains on expert-level scientific benchmarks like GPQA while matching the state-of-the-art on broader domain tasks, demonstrating a promising path toward agent systems that are simultaneously expert, adaptive, and efficient. We will release the code at https://github.com/myanvoos/ASpec.

## 1 Introduction

**Motivation.** The emergence of sophisticated multi-agent systems capable of tackling complex problems (Wu et al., 2024; Li et al., 2023; Hong et al., 2024) has marked a significant advance for autonomous agents. While effective, these foundational systems were often manually hand-crafted for specific tasks, which limited their scalability. In response, research has shifted towards automating aspects of these systems, starting with prompt optimization (Khattab et al., 2024; Yuksekgonul et al., 2025; Yang et al., 2024) or inter-agent communication via graph-based workflow representations (Zhuge et al., 2024; Liu et al., 2024; Zhang et al., 2025a), and then, to the designs of agent systems themselves. The automation of agent designs has since largely split into two distinct paradigms: task-level optimization and query-level adaptation. In the case of **(I) Task-Level Architecture Search**, prior works optimized for a single, static agent workflow for a specific task domain. These approaches, which mirror early approaches in AutoML and Neural Architecture Search (NAS) (Elsken et al., 2019), were pioneered by ADAS (Hu et al., 2025), which uses Meta Agent Search to iteratively program new agents in executable code; AFlow (Zhang et al., 2025b), which similarly adopts code representation but utilizes Monte Carlo Tree Search (MCTS) to efficiently navigate the search space; and AgentSquare (Shang et al., 2025), which employs module evolution and recombination to discover novel configurations in a constrained, modular code-based search space. The primary limitation of these methods is their intrinsic "one-size-fits-all" nature: by searching for a single best design for an entire task domain, they fundamentally lack the adaptability necessary to dynamically allocate inference resources or customize the structure for individual user queries.

To address the rigidity of task-level systems, a recent paradigm shift has focused on generating a unique workflow for each incoming query, **(II) Query-Level Architecture Adaptation**. MaAS (Zhang et al., 2025c) introduces the concept of an "agentic supernet", optimizing a probabilistic distribution of agent architectures during training and sampling a bespoke architecture from said distribution for each query during inference. This paradigm has been extended by other methods

---

[1]Corresponding author: myanvoos@gmail.com

like FlowReasoner (Gao et al., 2025), which uses a reasoning-based meta-agent to generate query-specific agent systems; ScoreFlow (Wang et al., 2025), which introduces Score-DPO, a method that fine-tunes its per-query workflow generator using quantitative evaluation scores; MAS-GPT (Ye et al., 2025), which trains an LLM to treat workflow construction as a generative task; and MAS-Zero (Ke et al., 2025), which employs a meta-agent at inference time to iteratively generate and refine agent configurations based on self-generated feedback. While these approaches offer superior adaptivity, they are challenged by the lack of long-term state. Because the architecture is regenerated or resampled for every query, the system incurs a significant "rediscovery" cost, and the individual components or agents are largely prevented from accumulating deep, persistent expertise over time.

The prior work demonstrate a critical chasm between monolithic, task-level robustness and adaptive, per-query regeneration. The former is static at inference, while the latter incurs "rediscovery" costs by repeatedly invoking meta-agents for architectural search in lieu of leveraging persistent knowledge, a system-level problem that a modular, agent-level memory addition would fail to address. Our proposed framework, ASPEC, reconciles these limitations by integrating the specialized mechanisms of self-evolving agents into a unified lifecycle within agent design automation. This lifecycle establishes stable, persistent agent archetypes deployed by a "retain-then-escalate" control policy, allowing the system to default to efficient *and* effective execution by relying on the persistent knowledge of its specialist agents.

**Contributions.** In short, our core contributions are as follows:

- We propose ASPEC, a framework that manages the full lifecycle of expert specialist agents via an automated two-stage methodology: **(I) Discovery**, where an LLM autonomously explores the design space of agent archetypes using evolutionary processes, and **(II) Cultivation**, where selected agents autonomously cultivate their expertise on a training corpus.
- We introduce "retain-then-escalate", a control policy that, instead of being either fully static or fully dynamic, defaults to retaining a stateful agent team across related queries to leverage expertise and minimize cost, only escalating to architectural resampling when needed.

**Related Work.** The mechanisms for autonomous discovery and expertise cultivation as seen in self-evolving agents have been explored individually across various research efforts. For instance, parallel to workflow optimization, a distinct stream of research has explored agent specialization via prompt optimization, starting with role assignment via ExpertPrompting (Xu et al., 2025), PromptBreeder (Fernando et al., 2023), and PromptAgent (Wang et al., 2024a). Multi-agent frameworks like EvoAgent (Yuan et al., 2025), which utilizes evolutionary algorithms to automatically generate and optimize multiple specialized agents with diverse settings and roles; MASS (Zhou et al., 2025), which optimizes individual role prompts alongside refining inter-agent communication; and AgentVerse (Chen et al., 2024a) and AutoAgents (Chen et al., 2024b), which dynamically synthesize and coordinate teams of expert roles, validate a critical insight: the *identity* of the agents is as important as their interaction topology. However, this specialization is often stateless, and the focus remains on generating an optimal team for a single task. In contrast, ASPEC's Discovery process generates persistent specialists whose structures are specifically designed to be retained and cultivated over time rather than generated for transient collaboration or discarded after a single optimization run.

Another stream of research in self-evolving agents is expertise cultivation, focused on endowing agents with non-parametric state (memory and experience) that persists beyond a single task interaction. Such mechanisms are embodied by works like Reflexion (Shinn et al., 2023), which allows agents to record natural-language critiques of their past actions in episodic memory to guide future behavior and avoid recurring mistakes, and Self-Refine (Madaan et al., 2023), which employs a continuous iterative refinement loop where the agent critiques and revises its initial outputs. Furthermore, ExpeL (Zhao et al., 2024) processes past trajectories to generate insights and rules to guide further interactions, AutoGuide (Fu et al., 2024) automatically generates context-aware guidelines from offline experiences, facilitating the provision of relevant knowledge for active decision-making processes, while Agent Workflow Memory (Wang et al., 2024b) records common subtask sequences that can be retrieved and reused without re-planning from scratch. These prior works illustrate how experiential knowledge can be accumulated and generalized into long-term competence.

While memory systems and reflection mechanisms exist, ASPEC proposes a systematic, two-stage lifecycle framework where the Cultivation phase is explicitly linked to the output of the Discovery

phase. This linkage ensures that the stateful expertise (memory/reflections) is accumulated within the designated, persistent specialist archetypes, facilitating the emergence of role-specific expertise.

## 2 PRELIMINARIES

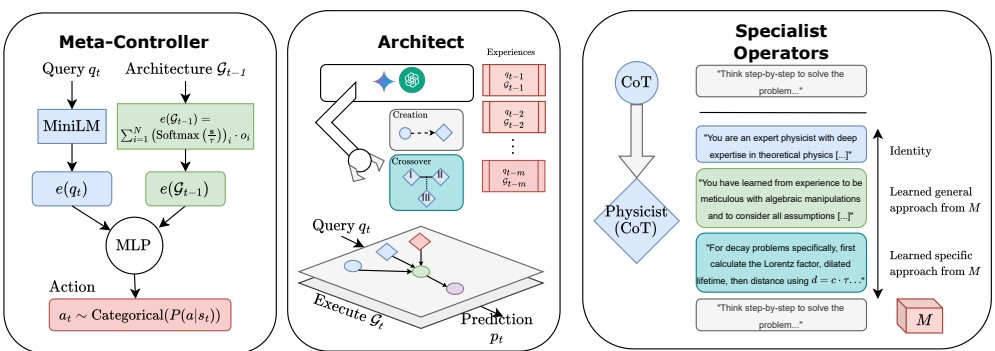

Figure 1: The three main components of ASPEC.

ASPEC can be framed as a Hierarchical Reinforcement Learning (HRL) methodology consisting of a low-level generative process for architectural redesign and agentic operator pool evolution, as well as a lightweight, high-level policy that learns *when* to invoke this process efficiently. We formally define these components below, starting with the modular units they operate upon: agentic operators.

**Definition (Agentic Operator).** Following MaAS (Zhang et al., 2025c), we define an agentic operator $O$ as a tuple $O = (\mathcal{M}, \mathcal{P}, \{\mathcal{T}_i\}_{i=1}^n)$ where $\mathcal{M} \in \mathbb{M}$ denotes the LLM backbone, $\mathcal{P} \in \mathbb{P}$ denotes the prompt, and $\{\mathcal{T}_i\} \subseteq \mathbb{T}$ denotes the available tools. A multi-agent system is then represented as a directed acyclic graph $\mathcal{G} = \{V, E\}$ where each vertex $v \in V$ represents an instance of an agentic operator and each edge $e \in E$ defines the connection between two operators.

To facilitate the evolutionary process at the heart of our methodology, we structure the operator pool $\mathbb{O}_t$ into two functionally distinct sets. First, the base operators ($\mathbb{O}_{\text{base}}$), a static set of foundational, stateless operators consisting of extensible single-/multi-agent systems, for instance Chain-of-Thought (Wei et al., 2022) or LLM-Debate (Du et al., 2024a). Second, the specialist operators ($\mathbb{O}_{\text{spec}}$), a dynamic set of operators derived from base operators.

A specialist $O_i^S \in \mathbb{O}_{\text{spec}}$ extends a base operator $O_i \in \mathbb{O}_{\text{base}}$ with a learned identity and a persistent memory while inheriting its foundational reasoning structure (e.g., "think step-by-step"). It is a tuple $O_i^S = (O_i, \mathcal{P}_s, M)$ where $\mathcal{P}_s$ is a specialized prompt and $M$ is a persistent, experience-driven memory module. We decompose $\mathcal{P}_s$ into an **identity**, which is a rich descriptor of who the agent is (Xu et al., 2025), and a set of **directives**, which are methodological principles for the agent's thought process, allowing for a rich and diverse "genetic" space of reasoning approaches (Naik et al., 2024).

**Definition (Architect).** The architect is the low-level generative component responsible for evolving the operator pool and redesigning the multi-agent architecture, implemented as an in-context learning LLM that operates via a multi-turn iterative reasoning process. We provide the prompt in Appendix G.1 and give an example of its reasoning in Appendix A.2. Functionally, given a query $q_t$, the Architect is a process $f_{\mathbb{A}}$ that maps a rich contextual input to a new system configuration

$$f_{\mathbb{A}}(q_t, \mathcal{H}_{t-m:t-1}, \mathbb{O}_{t-1}, \mathcal{G}_{t-1}) \rightarrow (\mathcal{G}_t, \mathbb{O}_t) \tag{1}$$

where $\mathcal{H}_{t-m:t-1}$ is a sliding window of the past $m$ experiences including the executed architectures and performance outcomes; $\mathbb{O}_{t-1}$ is the previous operator pool; and $\mathcal{G}_{t-1}$ is the current architecture. Its objective is to find an architecture that maximizes the immediate cost-aware utility while being general enough to be potentially retained for future tasks. We define this value in terms of the utility with respect to the oracle $a_t$, $U_t = U(\mathcal{G}_t; q_t, a_t)$, and the total costs of all API LLM calls, $C_t(\mathcal{G}_t)$.

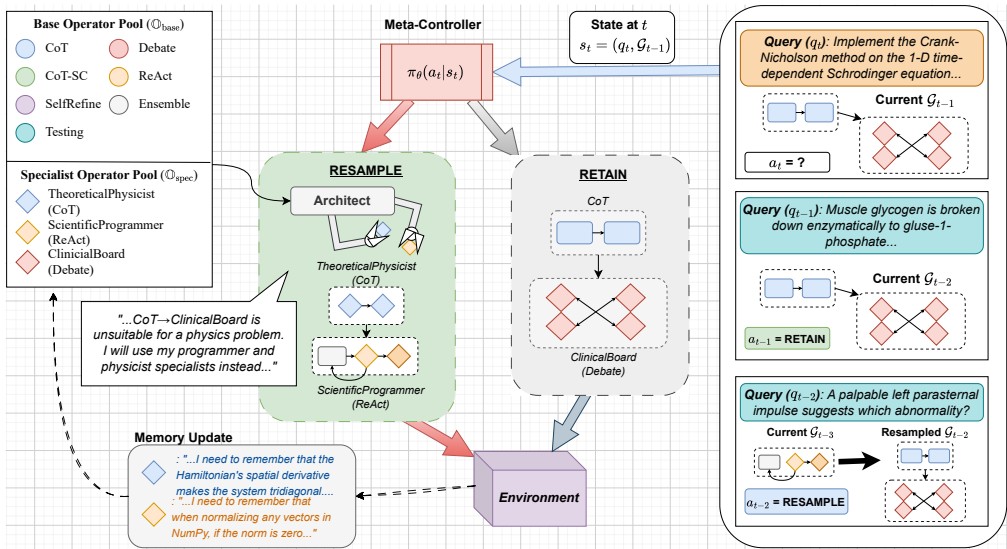

Figure 2: The online adaptation loop of ASPEC.

$$\mathcal{G}_t^* = \arg\max_{\mathcal{G}_i \in \mathcal{G}} \mathbb{E}\left[U_t - \lambda C_t(\mathcal{G}_t) + V_{\pi_\theta}(s_{t+1})\right] \tag{2}$$

where $V_{\pi_\theta}(s_{t+1})$ is the expected future value given the next state, formally defined in Equation 3. While this generative process enables adaptation, by continuously rebuilding the architecture, the system potentially forgoes the chance for the active specialists to deepen their expertise on the novel task. Additionally, and perhaps even more importantly, the Architect's invocation is computationally expensive and poses a practical challenge at scale. To address the trade-off between adaptability, experiential learning, and cost-efficiency, we propose the meta-controller, a lightweight gating module that decides when to escalate to the Architect during deployment.

**Definition (Meta-Controller).** The meta-controller is a neural policy $\pi_\theta(a_t|s_t)$ that makes a single high-level decision: retain the current agent architecture, or resample a new one for a given query. Its action space is discrete, that is, $\mathcal{A} = \{a_{\text{RETAIN}}, a_{\text{RESAMPLE}}\}$. We formulate the training of the meta-controller as a Markov Decision Process (MDP), where the action taken at step $t-1$ determines the architecture $\mathcal{G}_{t-1}$ available in the subsequent state $s_t$. The state $s_t$ at timestep $t$ is therefore:

$$s_t = (e_q(q_t), e_g(\mathcal{G}_{t-1})) \tag{3}$$

where $e_q(\cdot)$ and $e_g(\cdot)$ are fixed-length query and textual graph embeddings, embedded with MiniLM (Wang et al., 2020). While previous work (Zhang et al., 2025a) has used Graph Neural Networks (GNNs) to encode architectural topology, we opt for a simpler, query-aware semantic representation. Our *'bag-of-operators'* approach represents an architecture as an attention-weighted average of the embeddings of its constituent operators. The attention weights are computed based on the similarity between each operator and the input query embedding $e_q(q_t)$. This method, inspired by Vaswani et al. (2017), yields a dynamic, query-contextual state representation that captures *what* an architecture can do for a specific query without the significant training overhead of a dedicated GNN.

The explicit objective for the meta-controller is to maximize the expected discounted sum of future rewards over a stream of queries:

$$\pi_\theta^* = \arg\max_{\pi_\theta} \mathbb{E}\left[\sum_{t=0}^{t=T} \gamma^t \cdot R_t(s_t, a_t)\right], \quad \gamma \in [0, 1) \tag{4}$$

## 3 METHODOLOGY

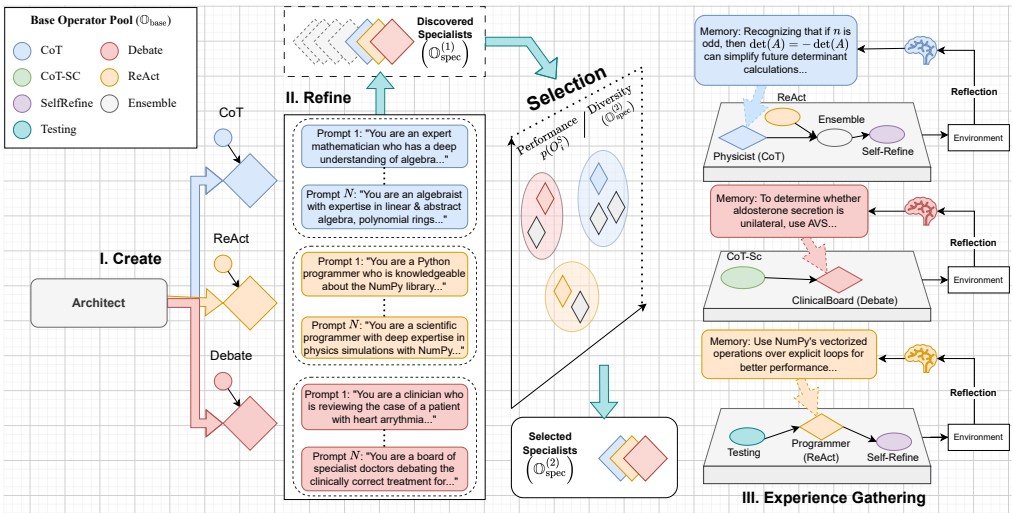

Figure 3: The offline automated specialist discovery and cultivation process.

Our framework's methodology is twofold. First, an end-to-end offline process discovers stateful specialists and trains the meta-controller (Figure 3 and Algorithm 2). These components are then deployed in an online adaptation loop to handle unseen queries, with the operator pool fixed (Figure 2 and Algorithm 1). To explore the space of possible specialists and identify a set of specialist operators $\mathbb{O}_{\text{spec}}$ such that the resulting operator pool is (1) high-performing, (2) diverse, and (3) specialized to the problem task domain **without human intervention**, we split the learning objectives into two distinct phases: an initial exploratory specialist discovery phase to address (1) and (2), and a focused, experience-gathering cultivation phase to address (3), mirroring how a human expert might first learn broad concepts and then deepen their knowledge through practice.

### 3.1 SPECIALIST DISCOVERY

Depicted as stages **I** and **II** in Figure 3, during the specialist discovery phase, the Architect iteratively evolves a pool of specialists using its full action space (detailed in Appendix G.1). We formalize the action space using the notions of creation and crossover.

**Creation.** Let $\mathbb{O}_{\text{spec}}^{(1)}$ be the pool of specialist operators during the specialist discovery phase and $\mathbb{O}_{\text{spec}}^{(2)}$ be the pool of specialist operators during the cultivation phase. For a query $q_t$, the Architect can propose a specialist $O_i^S \in \mathbb{O}_{\text{spec}}^{(1)}$ derived from a base operator $O_i$ by instantiating its prompt with a structured identity-directive pair. The creation process employs multi-variant synthesis with LLM adjudication. In practice, we overgenerate $S = 3$ candidate identity-directive variants with diverse pairs, then judge variants via an LLM-guided evaluation process that considers the reasoning methodology and domain coverage. We provide the prompts for the Judge in Appendix G.3.

To prevent early fragmentation, we enforce a dynamic pool size limit of $2 \times k$, where $k$ is the maximum size of the final specialist selection pool. If the pool exceeds this limit, the Architect is restricted from creating new specialists and must combine or prune existing agents, forcing the consolidation of narrow capabilities.

**Crossover.** Given parent specialist operators $O_1^S$ and $O_2^S$, the Architect can synthesize a child specialist $O_c^S$, similarly by using variant generation. This similarly triggers a multi-variant synthesis process with LLM adjudication that combines both parents' specialist identities and directives, preserving their expertise. We provide the prompts used to perform this synthesis in Appendix G.2.

**Selection.** At the end of the specialist discovery phase, we select the top-$k$ specialists for cultivation by solving a multi-objective optimization problem that balances performance and diversity:

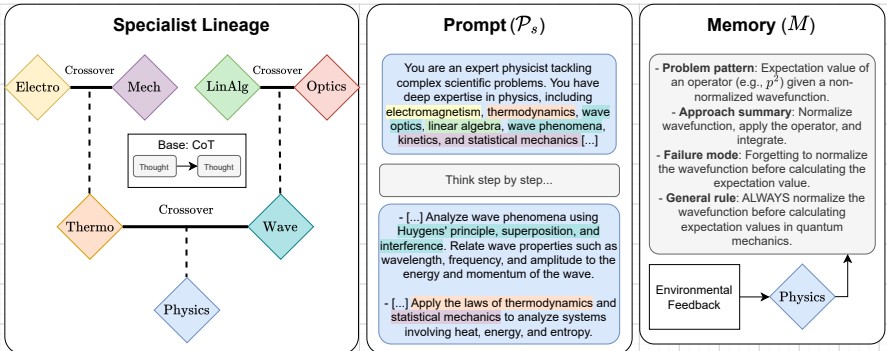

Figure 4: Case study of a physics specialist discovered on GPQA. The crossover action allows us to trace back the agent's "lineage" and identify aspects of its prompt that have been inherited from its ancestors. The full final prompt and more examples of its memory entries are in Appendix A.3.

$$\mathbb{O}_{\text{spec}}^{(2)} = \arg\max_{|\mathbb{O}_{\text{spec}}| \leq k} \left\{ \sum_{O_i^S \in \mathbb{O}_{\text{spec}}^{(1)}} p(O_i^S) + \text{Diversity}(\mathbb{O}_{\text{spec}}) \right\}$$

$$\text{Diversity}(\mathbb{O}_{\text{spec}}^{(2)}) = \sum_{j=1}^{k} \max_{O_i^S \in C_j \cap \mathbb{O}_{\text{spec}}} p(O_i^S)$$

(5)

where $p(O_i^S)$ represents the average performance of specialist $O_i^S$ and $C_j$ is the $j$-th cluster in embedding space obtained via K-means clustering on specialist operator embeddings.

## 3.2 SPECIALIST CULTIVATION

Depicted as stage **III** of Figure 3, during the specialist cultivation phase, the selected top-$k$ discovered specialists deepen their domain expertise through post-execution reflection on a training corpus. The cultivation process is applied independently to each specialist, resulting in distinct, specialized memories, as can be seen in Figure 4. For each specialist $O_i^S$ with accumulated memory $M_i$, we implement a semantic retrieval mechanism (Lewis et al., 2020) to inject relevant experience during tasks. Given a query $q_t$, we partition the memory into structured chunks, then inject the most relevant chunks as contextual knowledge during specialist execution.

## 4 RESULTS

**Benchmarks & Baselines.** We evaluate ASPEC on five public benchmarks across three domains: **math reasoning** with MATH (Hendrycks et al., 2021), **question answering** with MMLU (Hendrycks et al., 2021) and GPQA (Rein et al., 2024), **code generation** with HumanEval (Du et al., 2024b) and SciCode (Tian et al., 2024). In particular, GPQA and SciCode are expert-level QA and coding benchmarks respectively. Further details on the dataset statistics are in Appendix F.

We select 13 representative baselines across **(1) hand-designed single agents**, in particular Chain-of-Thought (Wei et al., 2022), Self-Refine (Madaan et al., 2023), Self-Consistency (Wang et al., 2023), Reflexion (Shinn et al., 2023); **(2) hand-designed multi-agents**, in particular LLM-Debate (Du et al., 2024a), DyLAN (Liu et al., 2024); **(3) automated agent specialisation methods** with Role Assignment (Xu et al., 2025), AutoAgents (Chen et al., 2024b), EvoAgent (Yuan et al., 2025); and **(4) autonomous agent design frameworks**, including query-level MaAS (Zhang et al., 2025c), and task-level AFlow (Zhang et al., 2025b) and ADAS (Hu et al., 2025). Details for the baseline setups are in Appendix B.

**Implementation.** We select Gemini 2.0 Flash to be the standard execution model across all methods, alongside GPT-4o-mini and Llama 3.3 70B Instruct in Figure 5. We set the size of the sliding window in Equation 1 to be $m = 10$ and the maximum number of specialists in Equation 5 to be $k = 5$.

Table 1: Performance comparison across methods. We use Gemini 2.0 Flash with a temperature of $T = 0.3$ consistently across all methods. Best results are in **bold**, second-best are underlined.

| Method | MATH | HumanEval | MMLU | GPQA | SciCode (SP) | Average |
|---|---|---|---|---|---|---|
| Vanilla | 73.2 | 87.8 | 86.0 | 56.3 | 24.0 | 65.3 |
| CoT (Wei et al., 2022) | 74.5 | 90.4 | 88.2 | 58.2 | 24.3 | 65.5 |
| CoT-SC (Wang et al., 2023) | 75.1 | 91.2 | 88.8 | 57.1 | 25.2 | 67.5 |
| Self-Refine (Madaan et al., 2023) | 74.8 | 91.3 | 88.5 | 57.4 | 24.6 | 67.3 |
| Reflexion (Shinn et al., 2023) | 73.5 | 86.8 | 88.5 | 57.1 | 25.1 | 66.2 |
| LLM-Debate (Du et al., 2024a) | 74.4 | 85.5 | 87.1 | 59.7 | 24.0 | 66.1 |
| DyLAN (Liu et al., 2024) | 75.4 | 89.3 | 88.9 | 61.3 | 25.2 | 68.0 |
| Role Assignment (Xu et al., 2025) | 72.4 | 91.2 | 89.5 | 57.4 | 23.5 | 67.6 |
| AutoAgents (Chen et al., 2024b) | 73.4 | 88.0 | 85.3 | 56.8 | 24.8 | 65.7 |
| EvoAgent (Yuan et al., 2025) | 75.9 | 90.2 | 88.3 | 61.5 | 24.8 | 68.1 |
| ADAS (Hu et al., 2025) | 74.5 | 82.9 | 90.0 | 58.2 | 24.8 | 66.2 |
| AFlow (Zhang et al., 2025b) | 76.5 | 89.3 | **90.5** | 61.3 | 24.3 | 68.4 |
| MaAS (Zhang et al., 2025c) | 74.4 | **91.6** | 87.3 | 57.8 | 25.6 | 67.4 |
| ASPEC | **77.3** | 91.4 | 90.0 | **62.8** | **26.6** | **69.6** |

Figure 5: Cross-model (left) and cross-benchmark (right) transferability results. We evaluate both the full ASPEC and ASPEC with *only* specialists trained on a different benchmark.

| LLM Backbone | GPQA | MATH | HumanEval |
|---|---|---|---|
| Gemini 2.0 Flash | 56.3 | 73.2 | 87.8 |
| ASPEC (Gemini 2.0 Flash) | 62.5 | 77.3 | 91.4 |
| GPT-4o-mini | 38.2 | 61.8 | 86.6 |
| ASPEC (GPT-4o-mini) | 43.8 | 64.7 | 90.9 |
| Llama 3.3 70B Instruct | 45.6 | 51.3 | 88.5 |
| ASPEC (Llama 3.3 70B Instruct) | 53.5 | 54.8 | 90.8 |

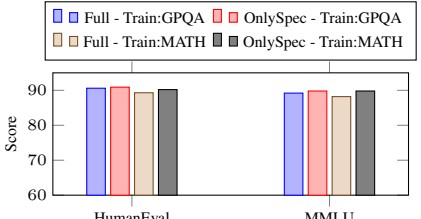

**Performance Analysis.** The results from Table 1 demonstrate that ASPEC can consistently match or outperform existing hand-crafted or automated agentic systems across mathematical reasoning, question answering, and coding. Its benefits are most pronounced on GPQA, where it achieves a score of 62.8%. This represents a substantial 6.5% improvement over the vanilla Gemini 2.0 Flash model. Furthermore, ASPEC surpasses the leading hand-designed agent (LLM-Debate) by 3.1%, the top autonomous agent framework (AFlow) by 1.5%, and the best automated agent specialisation method (EvoAgent) by 1.3%. ASPEC also leads on SciCode, a benchmark composed of realistic scientific research problems that are decomposed into sequential subproblems. We note that the "retain-then-escalate" structure allows retained specialists to build upon context and learned knowledge from previous steps, which is crucial for success in multi-part scientific coding.

This naturally leads to the question of whether specialists trained on specific domains can be transferred to other domains. To this end, Figure 5 shows that the performance gains from the ASPEC methodology are robustly transferable across different models and benchmarks. In the cross-benchmark analysis (Figure 5, right), we compare the standard configuration against an ablation labeled ONLYSPEC, where the operator pool is restricted exclusively to specialists trained on a different source domain (e.g., utilizing MATH-trained specialists for HumanEval), and find that the ONLYSPEC configuration matches or even slightly exceeds the performance of the full system.

We attribute this to the cultivation of "T-shaped" reasoning strategies for specialists (Appendix G.3); furthermore, restricting the pool prevents the Architect from defaulting to "safe" but less capable generalist base operators, effectively forcing the utilization of these expert reasoning archetypes.

Table 2: Efficiency comparison across methods on the GPQA benchmark.

| Method | Training | | | Inference | | | |
| --- | --- | --- | --- | --- | --- | --- | --- |
| | Total tokens | Total costs (USD) | Wall clock (min) | Total tokens | Total costs (USD) | Wall clock (min) | Accuracy (%) |
| CoT-SC (Wang et al., 2023) | – | – | – | 3,757,527 | **0.85** | 58 | 57.1 |
| LLM-Debate (Du et al., 2024a) | – | – | – | 4,081,114 | 0.94 | 50 | 59.7 |
| EvoAgent (Yuan et al., 2025) | – | – | – | 7,080,338 | 1.45 | 75 | 61.8 |
| AFlow (Zhang et al., 2025b) | 102,012,408 | 20.14 | 257 | 9,997,154 | 1.58 | **45** | 61.3 |
| MaAS (Zhang et al., 2025c) | 11,600,690 | 3.43 | 139 | 11,015,542 | 2.07 | 93 | 57.8 |
| ASPEC | **2,395,636** | **1.38** | **53** | **3,204,549** | 0.88 | 63 | **62.8** |

**Efficiency Analysis.** Table 2 demonstrates that ASPEC is cost-efficient across both training and inference. In particular, running the offline training process on GPQA incurred only a total cost of 1.38 USD. We find that once a strong specialist pool has been found, the Architect often prefers lean architectures utilizing those specialists. As shown in Table 6, removing specialists causes costs to increase significantly – the Architect becomes under-confident in its generalist pool and samples highly complex, but redundant multi-agent architectures in an attempt to compensate.

## 5 DISCUSSION

### 5.1 ABLATIONS OF SYSTEM COMPONENTS AND CONTROL POLICIES

Figure 6: Ablation study of our framework's components (left) and sensitivity to the maximum number of specialists $k$ and sliding window length $m$ (right) on GPQA. For sensitivity plots, the central line shows the mean performance over 4 runs.

| Method | Accuracy (%) | Total cost (USD) |
| --- | --- | --- |
| *Control Policy Alternatives* | | |
| ASPEC w/ random policy | 58.3 | 1.05 |
| ASPEC w/ $h = 0.2$ | 59.6 | 1.21 |
| ASPEC w/ LLM-as-gate (Gemini 2.0 Flash) | 62.5 | 3.74 |
| *System Components* | | |
| ASPEC w/o specialist operators | 57.4 | 2.26 |
| ASPEC w/o base operators | 61.3 | 0.48 |
| ASPEC w/o meta-controller | 62.7 | 2.0 |
| ASPEC w/o Architect | 61.0 | 1.28 |
| ASPEC w/o specialist memory | 61.4 | 0.94 |
| ASPEC | 62.8 | 0.88 |

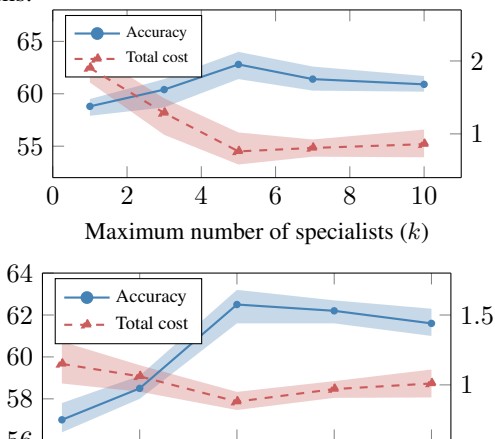

We perform an ablation study on five key components: **(I) without specialist operators**, with the operator pool restricted to $\mathbb{O} = \mathbb{O}_{\text{base}}$ for all $q_t$; **(II) without base operators**, with $\mathbb{O} = \mathbb{O}_{\text{spec}}$ for all $q_t$; **(III) without meta-controller**, which is akin to always resampling; and **(IV) without architect**, in which we construct a static architecture consisting of all specialist operators in $\mathbb{O}_{\text{spec}}$, and **(V) without specialist memory**. Furthermore, we perform additional experiments on a suite of alternative control policies. These include a random "coin-flip", a cosine similarity heuristic (resample if the cosine similarity of $q_t$ and $\mathcal{G}_{t-1}$ is below a threshold of $h = 0.2$), and LLM-as-gate.

As seen in Figure 6 (left), removing specialists causes a 5.4% drop in performance from 62.8% to 57.4% and a near tripling of the total cost from 0.88 USD to 2.26 USD. Additionally, removing base operators, but keeping specialists, resulted in a lesser 1.5% drop. This demonstrates that the expert specialists are the primary drivers of both performance and efficiency. Removing the meta-controller results in a comparable performance of 62.7% at a $\sim 2.3$ times increase in total costs. We note that

even in this mode, the Architect can learn to proxy the "retain" action by re-sampling $\mathcal{G}_{t-1}$, but this invocation process is fundamentally less efficient that the explicit "retain" decision made by the controller. Figure 6 (left) further reveals that the alternative control policies yield significantly lower accuracy at $58.3\%$ compared to the meta-controller's $62.8\%$, and while the LLM-as-gate policy achieves a comparatively high accuracy $62.5\%$, it does so at a substantially higher cost, $\sim 4.25$ times that of the meta-controller.

## 5.2 SENSITIVITY ANALYSES

We analyze the sensitivity of ASPEC to two main parameters: the maximum size of the specialist pool, $k$, from Equation 5, and the length of the sliding window from Equation 1, $m$. As shown in Figure 6, setting $k$ at both extremities reduced performance, suggesting a light Goldilocks-like effect on GPQA. At $k = 1$, the system achieves $58.8\%$, performing similarly to the "ASPEC w/o specialist operators" ablation as seen in Table 6, indicating that a single specialist lacks the domain coverage to outperform generalist operators. Conversely, at $k = 10$, performance drops to $60.9\%$, which aligns closely with the "ASPEC w/o specialist memory" ablation. We attribute this to experience fragmentation: with a larger pool size, individual specialists are selected less frequently by the Architect during the Cultivation phase. Since memory is only acquired upon execution, these "sparse" specialists fail to accumulate the dense history required to form deep expertise.

We hypothesize that this is not necessarily a hard limitation and the Goldilocks distribution might depend more on the depth of an average specialist's experience and exposure to problems.

## 5.3 CONVERGENCE OF THE SPECIALIST DISCOVERY PROCESS

Figure 7: Visualization of discovered specialist operator embeddings on a "narrow" domain benchmark (GPQA) and on a "broad" domain benchmark (MMLU).

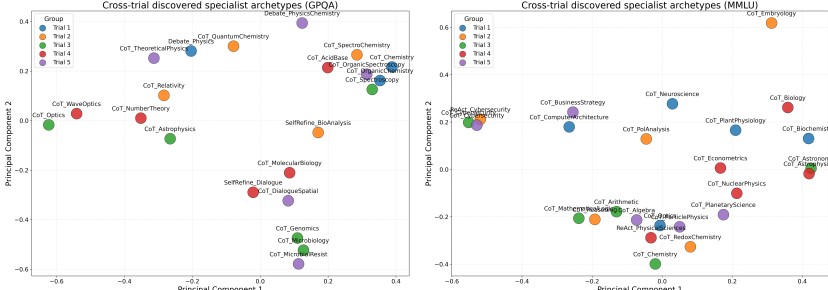

To determine whether ASPEC's discovery process reliably finds similar expert archetypes, we embedded the prompts of discovered specialists across 5 independent trials and plotted them in Figure 7. We find that there is strong convergence on GPQA (Figure 7, left), with different runs independently discovering the same key roles (chemistry, biology, physics), demonstrating the robustness of the process for specialized domains. Conversely, on the broad-domain MMLU benchmark (Figure 7, right), the process shows some divergence, exploring different but viable team compositions to cover the vast topic space. Even so, we find pockets of convergence in well-defined sub-domains like the physical sciences. Taken together, these results show that the ASPEC discovery process adapts its convergence/divergence behavior based on the specificity of the target domain.

### 5.3.1 RATIONALITY ANALYSIS.

We compare a learned meta-controller's decisions against the LLM-as-gate "oracle proxy" in Table 6. On GPQA, the controller learns a pragmatic economic policy, where its high rate of "overconfident" disagreements with the perfectionist oracle reflects a deliberate trade-off for cost efficiency. On MMLU, this behavior persists, but instances of "wasteful caution" reveal the limitations of its lightweight state representation, leading to unnecessary resampling.

Figure 8: Side-by-side comparison of confusion matrices for GPQA (left) and MMLU (right).

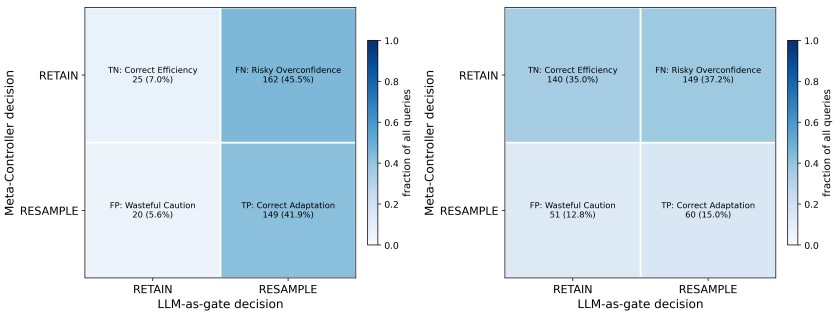

# 6 LIMITATIONS & FUTURE WORKS

A key future direction is the development of a rigorous theoretical framework to model the convergence properties of the specialist discovery process with respect to factors like domain breadth, potentially leading to principles for self-tuning the discovery process. Future work should also validate ASPEC's applicability in more diverse environments, particularly on complex, real-world software engineering tasks such as those in SWE-bench (Jimenez et al., 2023). Intuitively, specialists discovered and cultivated on a specific repository could autonomously internalize its unique conventions and APIs, a promising avenue for automating repository-specific expertise without manually engineered rules. Finally, specialists may amplify training biases through memory cultivation, a risk that warrants further investigation and the development of mitigation strategies.

While our lightweight meta-controller is crucial for efficiency, we identify its alignment with an "oracle proxy" LLM-as-gate policy as another critical area for improvement. The results of our ablation study on GPQA in Table 6 might be masking an underlying limitation: the meta-controller's decision-making process diverges from the oracle proxy's. This divergence can become a significant weakness when its lightweight state representation leads to errors such as unnecessary resampling or over-cautious retaining. The central challenge is to design a gating mechanism that achieves the decision-making fidelity of the LLM-as-gate oracle proxy while retaining the low computational overhead of a small, specialized policy.

Finally, we observe that the interplay between the meta-controller's policy, the Architect's choices, and the specialists' memory accumulation creates a complex, co-evolutionary dynamic. A conservative "Retain" policy concentrates experience into a smaller set of active architectures, potentially guiding those agents to develop broader, more resilient memories to cope with slightly mismatched queries. Conversely, a highly dynamic policy distributes experience more sparsely across the specialist pool. Furthermore, because the Architect conditions its decisions on recent history (sliding window), it may develop path-dependent preferences for certain teams that "suffice" even if they are not optimal, further influencing the distribution of experience. Future work could explicitly model this joint optimization to ensure the control policy and specialist cultivation are perfectly aligned.

# 7 CONCLUSION

This paper introduced ASPEC, a framework designed to bridge the gap between static, efficient agent workflows and adaptive, per-query optimizers. Our central contribution is a methodology for creating and managing stateful specialist agents that accumulate expertise over time, mirroring human learning. This is achieved through an automated lifecycle of evolutionary discovery and experiential cultivation, governed by a "retain-then-escalate" policy that ensures cost-effective adaptation. Our results on challenging scientific benchmarks such as GPQA suggest that this agent-centric approach can lead to substantial performance improvements without sacrificing efficiency. We believe this work presents a promising direction for autonomously creating agent systems that can develop deep expertise while retaining the flexibility to adapt to new challenges. [1]

---

[1]Large Language Models (LLMs) were used to assist in writing this paper.

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

# A CASE STUDY

## A.1 META-CONTROLLER DECISION-MAKING

We provide a few examples of a trained meta-controller's decision-making process on GPQA. These include **(I) rational decisions**, such as retaining or resampling sensibly, and **(II) irrational decisions**, when the imperfect meta-controller chooses to retain a mismatching architecture or resample a matching architecture, thereby incurring expensive, unnecessary costs from the Architect call.

### A.1.1 RATIONAL DECISIONS

```
Query: "Determine which set of states mentioned below are only entangled
    states:
(a) (1/ 30 )* (|00>+ 2i|01>    3|10>    4i|11>)
(b) (1/5)* (|00>+ 2i|01>    2|10>    4i|11>)
(c) (1/2)(|00>+ |01>+|10>    |11>)
(d) (1/2)(|00>+ |01>-|10>    |11>)."

Current architecture: [["CoT"], ["CoT_TheoreticalPhysics"]]
Action taken: "RETAIN"
Resulting architecture: [["CoT"], ["CoT_TheoreticalPhysics"]]
Outcome: CORRECT
```

```
Query: "Identify the missing reagents in the following reaction.
(3r,5r,7r)-adamantane-1-carboxylic acid + A --->
    (3r,5r,7r)-adamantane-1-carbonyl azide + B --->
    (3s,5s,7s)-adamantan-1-amine."

Current architecture: [["CoT"], ["CoT_TheoreticalPhysics"]]
Action taken: "RESAMPLE"
Resulting architecture: [["CoT_OrganicSpectroscopy"]]
Outcome: CORRECT
```

### A.1.2 IRRATIONAL DECISIONS

```
Query: "The Cope rearrangement is a chemical reaction where a 1,5-diene
    molecule undergoes rearrangement, resulting in a change in the
    positions of its carbon-carbon double bonds. This rearrangement can
    be initiated by heat or light and is valuable for creating complex
    organic compounds with changed structures. Select the major products
    from the following rearrangements [...]"

Current architecture: [["CoT"], ["CoT_TheoreticalPhysics"]]
Action taken: "RETAIN"
Resulting architecture: [["CoT"], ["CoT_TheoreticalPhysics"]]
Outcome: INCORRECT
```

```
Problem index: 2
Query: "Astronomers are searching for exoplanets around two stars with
    exactly the same masses. Using the RV method, they detected one
    planet around each star, both with masses similar to that of Neptune
    [...]

The question is: How many times is the orbital period of planet #2
    longer than that of planet #1?"

Current architecture: [["CoT_TheoreticalPhysics"]]
Action taken: "RESAMPLE"
Resulting architecture: [["CoT_TheoreticalPhysics"]]
Outcome: CORRECT
```

## A.2 ARCHITECT REASONING PROCESS

---

**Figure 9: A three-turn reasoning process of the Architect.**

**Initial State**    During the discovery phase, the system receives a query that asks it to solve a complex quantum mechanics problem involving both wave function normalization and energy eigenvalue calculations. The current architecture is `[[CoT]]`. The operator pool contains basic operators and a `CoT_Astrophysics` specialist, but lacks quantum mechanics expertise.

- - - - - - - - - - - - - - - - - - - - - - - - - - - - - - - - - - - - - - - - - - - - - - -

**Turn 1: Specialist Creation**    The Architect analyzes the query and identifies the need for domain-specific expertise in quantum mechanics.

```
<action>CREATE_OPERATOR</action>
<reasoning>
The query requires deep expertise in quantum mechanics,
    specifically wave function normalization and energy eigenvalue
    calculations. I will create a specialized operator that can
    handle the mathematical rigor required for quantum mechanical
    problems.
</reasoning>
<new_operator_name>CoT_QuantumMechanics</new_operator_name>
<base_template>CoT</base_template>
<identity>You are an expert quantum physicist specializing in wave
    function analysis and energy eigenvalue problems. You have deep
    expertise in Schrodinger equations, Hilbert spaces, and
    operator theory, with broad knowledge across theoretical
    physics and mathematical methods.</identity>
<directives>
    - Always verify wave function normalization conditions
    - Apply systematic eigenvalue analysis using appropriate
        boundary conditions [...]
</directives>
```

- - - - - - - - - - - - - - - - - - - - - - - - - - - - - - - - - - - - - - - - - - - - - - -

**Turn 2: Operator Combination**    After creating the quantum specialist, the Architect decides to combine it with an existing physics specialist.

```
<action>COMBINE_OPERATORS</action>
<reasoning>
Analyzing the recent sliding window of past experiences reveals a
    pattern of diverse physics problems including both quantum
    mechanics and astrophysics queries. Rather than maintaining
    narrow specialists that may underperform on this broader
    distribution, I will combine CoT_QuantumMechanics and
    CoT_Astrophysics into a unified CoT_TheoreticalPhysics
    specialist.
</reasoning>
<target_operator_1>CoT_QuantumMechanics</target_operator_1>
<target_operator_2>CoT_Astrophysics</target_operator_2>
<new_operator_name>CoT_TheoreticalPhysics</new_operator_name>
<identity>You are an expert theoretical physicist with deep
    expertise in quantum mechanics and astrophysics. You combine
    rigorous mathematical analysis with physical intuition to solve
    complex problems in quantum theory, astrophysics, and
    mathematical physics.</identity>
<directives>
- For quantum mechanics problems, always verify wave function
    normalization conditions and apply systematic eigenvalue
    analysis [...]
- For astrophysics problems, apply scaling analysis and use
    dimensional arguments to constrain solutions [...]
```

```
</directives>
```

**Turn 3: Architecture Finalization**   With the enhanced specialist available, the Architect designs the optimal processing pipeline.

```
<action>FINALIZE_ARCHITECTURE</action>
<reasoning>
I will create a two-layer architecture:
    [['CoT_TheoreticalPhysics'], ['SelfRefine']]. The theoretical
    physics specialist will provide the initial detailed analysis,
    and SelfRefine will critically evaluate and improve the
    solution.
</reasoning>
<architecture>[["CoT_TheoreticalPhysics"],
    ["SelfRefine"]]</architecture>
```

## A.3  ANATOMY OF A SPECIALIST

We provide an example of a physicist that has been autonomously discovered and culti­vated on GPQA. This specialist was the result of a crossover between two parent special­ists, CoT_THERMODYNAMICS and CoT_WAVEOPTICS, who themselves descended from CoT_ELECTROMAGNETISM, CoT_LINEARALGEBRA, CoT_OPTICS, and CoT_MECHANICS.

---

**Specialist Prompt: CoT_PHYSICS**

You are an expert physicist tackling complex scientific problems. You have deep expertise in physics, including electromagnetism, thermodynamics, wave optics, linear algebra, wave phenomena, kinetics, and statistical mechanics. When faced with a complex problem, you always start by identifying the fundamental physical principles at play, breaking down the problem into its core components before attempting to solve it. You visualize physical phe­nomena as interconnected networks of energy and momentum, allowing you to intuitively understand their behavior.

Think step by step and derive a concise final answer.

- Focus on identifying the fundamental physical principles underlying the problem.

- Apply knowledge from various areas of physics, including electromagnetism, ther­modynamics, kinetics, wave optics, linear algebra. Consider the interplay between physics, chemistry, and biology when relevant.

- Prioritize dimensional analysis and order-of-magnitude estimates to quickly assess the plausibility of different solutions. Likewise, simplify complex problems by identifying dominant terms and making appropriate approximations.

- Analyze wave phenomena using Huygens' principle, superposition, and interference. Relate wave properties such as wavelength, frequency, and amplitude to the energy and momentum of the wave. Apply the laws of thermodynamics and statistical mechanics to analyze systems involving heat, energy, and entropy.

**# Learned from experience:**
- Prioritize accurate identification of fundamental transformations (e.g., electron flow) before making broader predictions.

---

- When comparing results from different methodologies, explicitly consider the limitations and biases inherent in each technique. Focus on underlying mechanisms and principles rather than superficial alignment of results.

- Consider frequency and averaging effects when integrating data from population-level and single-entity measurements.

---

**Specialist Memory: COT_PHYSICS**

**# Structured memory entry:**

- Problem pattern: EM wave attenuation; inconsistent parameters lead to physically impossible results (e.g., amplification instead of attenuation).

- Approach summary: Verify problem consistency by calculating attenuation from given parameters. Identify and state inconsistencies explicitly.

- Failure mode: Blindly applying formulas without checking physical plausibility; incorrect assumptions about attenuation contributions.

- General rule: Before solving, check if given parameters yield physically plausible results. If not, state the flaw and assumptions made for a solution.

**# Structured memory entry:**

- Problem pattern: Expectation value of an operator (e.g., $p^2$) given a non-normalized wavefunction.

- Approach summary: Normalize wavefunction, apply the operator, and integrate.

- Failure mode: Forgetting to normalize the wavefunction before calculating the expectation value.

- General rule: ALWAYS normalize the wavefunction before calculating expectation values in quantum mechanics.

**# Structured memory entry:**

- Problem pattern: Particle decay (e.g., $\Pi \rightarrow \mu + \nu$) with known rest masses and initial state. Find KE of products.

- Approach summary: Apply energy and momentum conservation. Use relativistic energy-momentum relation ($E^2 = (pc)^2 + (mc^2)^2$) to relate KE and momentum.

- Failure mode: Incorrectly applying relativistic formulas or conservation laws; algebraic errors in solving the equations.

- General rule: In particle decay, use energy/momentum conservation and relativistic relations. If one particle is at rest initially, simplify accordingly.

---

# B  BASELINES

In this section, we detail the implementation for each of the baseline methods. For Chain-of-Thought (Wei et al., 2022), Self-Consistency (Wang et al., 2023), Self-Refine ((Madaan et al., 2023)), and LLM-Debate (Du et al., 2024a), we refer to Appendix C for the configuration details, as they were used as seed base operators in ASPEC. For Reflexion, we adhere to the implemention provided in (Shinn et al., 2023). Following ADAS (Hu et al., 2025), we implement Role Assignment (Xu et al., 2025) by prompting a role-selector LLM to choose a role from a predefined set, then use another LLM to act as the chosen role to answer the question.

For each of the benchmarks, the roles for Role Assignment were:

- **MATH**: Algebraist, Number Theorist, Real Analyst, Statistician, Logician
- **HumanEval**: Senior Python Engineer, Algorithms Expert, Software Architect, Data Scientist, Competitive Programmer
- **MMLU**: Biologist, Physicist, Mathematician, Engineer, Doctor, Lawyer
- **GPQA**: Physicist, Chemist, Biologist, Scientific Reasoning Expert, Graduate Student
- **SciCode**: Biologist, Physicist, Chemist, Computer Scientist, Mathematician

For DyLAN and EvoAgent, we directly used the implementations from Liu et al. (2024) and (Yuan et al., 2025). We adhered to the official configuration for AutoAgents (Chen et al., 2024b). For ADAS (Hu et al., 2025), we set the Meta Agent Search's $n$-generation to 20. For MaAS, our experimental setup directly utilized the optimized graphs and operator spaces from (Zhang et al., 2025c) for MATH and HumanEval. For benchmarks not explicitly included in the MaAS repository (GPQA, MMLU, SciCode), we implemented the operator space as described in the appendix. Following Zhang et al. (2025c), for AFlow, we utilized Gemini 2.0 Flash consistently throughout our experiments in place of GPT-4o-mini and Claude 3.5 Sonnet for homogeneity.

## C  OPERATOR SPACE

Following MaAS (Zhang et al., 2025c), we use the following operator space for our base operators:

- **Chain-of-Thought** (Wei et al., 2022), which encourages the execution LLM to think step-by-step before outputting an answer.
- **ReAct** (Yao et al., 2023), allowing the execution LLM to use a library of tools to answer the question.
- **Self-Consistency** (Wang et al., 2023), which aggregates five Chain-of-Thought answers and majority votes to agree on a final answer.
- **Self-Refine** (Madaan et al., 2023), which iteratively refines an initial Chain-of-Thought answer over five iterations.
- **LLM-Debate** (Du et al., 2024a), which uses multiple execution LLMs to debate against each other to reach a final consensus. We similarly use three debaters and two rounds of debate in our implementation.
- **Ensemble** (Jiang et al., 2023), which takes in two or more answers from different sources and uses pairwise ranking to aggregate these responses into a final answer.
- **Testing** (Huang et al., 2024), which generates test cases for subsequent execution LLMs given a coding problem.

## D  ALGORITHMS

---

**Algorithm 1:** Online adaptation algorithm of ASPEC

---

**Input:** Trained meta-controller $\pi_\theta$; operator pool $\mathbb{O}$; queries $Q = \{q_1, \ldots, q_T\}$; sliding window buffer $\mathcal{H}$.

Initial graph $\mathcal{G}_0$.

**for** $t = 1, 2, \ldots, T$ **do**
    Construct state $s_t = (e_q(q_t), e_g(\mathcal{G}_{t-1}))$;
    Sample action $a_t \sim \pi_\theta(a_t|s_t)$;
    **if** $a_t = a_{RESAMPLE}$ **then**
        $\mathcal{G}_t \leftarrow f_{\mathbb{A}}(q_t, \mathcal{H}_{t-m:t-1}, \mathbb{O}, \mathcal{G}_{t-1})$;
    **else**
        $\mathcal{G}_t \leftarrow \mathcal{G}_{t-1}$;
    **end**
    $p_t \leftarrow \text{Execute}(\mathcal{G}_t, \mathbb{O}, q_t)$;
    $U_t, C_t \leftarrow \text{Evaluate}(p_t, a_t)$;
    Store experience $(q_t, \mathcal{G}_t, S_t, C_t)$ in $\mathcal{H}$;
**end**

---

---

**Algorithm 2:** Offline specialist discovery and cultivation

---

**Input:** Queries $Q = \{q_1, \ldots, q_T\}$; base operator set $\mathbb{O}_{\text{base}}$.

Initial operator pool $\mathbb{O}_0 = \mathbb{O}_{\text{base}}$, initial specialist pool $\mathbb{O}_{\text{spec}}^{(0)} \leftarrow \emptyset$, random-weights

meta-controller $\pi_\theta^{(0)}$; empty sliding window buffer $\mathcal{H} \leftarrow \emptyset$.

**for** $t = 1, 2, \ldots, T$ **do**

    Construct state $s_t = (e_q(q_t), e_g(\mathcal{G}_{t-1}))$;

    Sample action $a_t \sim \pi_\theta(a_t|s_t)$;

    **if** $a_t = a_{RESAMPLE}$ **then**

        $a_{\mathbb{A}} \leftarrow f_{\mathbb{A}}(q_t, \mathcal{H}_{t-m:t-1}, \mathbb{O}_{t-1}, \mathcal{G}_{t-1})$

        **if** $a_{\mathbb{A}} = CREATE\_OPERATOR$ **then**

            $O_{\text{new}} \leftarrow \text{CreateSpecialist}(q_t, \mathbb{O}_{\text{base}})$ ;

            $\mathbb{O}_{\text{spec}}^{(t)} \leftarrow \mathbb{O}_{\text{spec}}^{(t-1)} \cup \{O_{\text{new}}\}$ ;

            $\mathbb{O}_t \leftarrow \mathbb{O}_{t-1} \cup \mathbb{O}_{\text{spec}}^{(t)}$

        **end**

        **else if** $a_{\mathbb{A}} = COMBINE\_OPERATOR$ **then**

            $(O_1, O_2) \leftarrow \text{SelectOperators}(\mathbb{O}_{\text{spec}}^{(t-1)})$ ;

            $O_{\text{child}} \leftarrow \text{Combine}(O_1, O_2, q_t)$ ;

            $\mathbb{O}_{\text{spec}}^{(t)} \leftarrow (\mathbb{O}_{\text{spec}}^{(t-1)} \setminus \{O_1, O_2\}) \cup \{O_{\text{child}}\}$ ;

            $\mathbb{O}_t \leftarrow \mathbb{O}_{t-1} \cup \mathbb{O}_{\text{spec}}^{(t)}$

        **end**

        **else if** $a_{\mathbb{A}} = PRUNE\_OPERATOR$ **then**

            $O_{\text{to\_prune}} \leftarrow \text{SelectOperator}(\mathbb{O}_{\text{spec}}^{(t-1)})$ ;

            $\mathbb{O}_{\text{spec}}^{(t)} \leftarrow \mathbb{O}_{\text{spec}}^{(t-1)} \setminus \{O_{\text{to\_prune}}\}$ ;

            $\mathbb{O}_t \leftarrow \mathbb{O}_{t-1} \cup \mathbb{O}_{\text{spec}}^{(t)}$

        **end**

        $\mathcal{G}_t \leftarrow f_{\mathbb{A}}(\mathcal{H}_{t-m:t-1}, \mathbb{O}_t, \mathcal{G}_{t-1})$ ;

    **else**

        $\mathcal{G}_t \leftarrow \mathcal{G}_{t-1}$ ;

    **end**

    $p_t \leftarrow \text{Execute}(\mathcal{G}_t, \mathbb{O}_t, q_t)$;

    $U_t, C_t \leftarrow \text{Evaluate}(p_t, a_t)$;

    $\pi_\theta^{(t)} \leftarrow \text{UpdateWeights}\left(U_t, C_t, a_t, \pi_\theta^{(t-1)}\right)$

    **forall** $O \in SpecialistsUsedIn(\mathcal{G}_t, \mathbb{O}_t)$ **do**

        $r \leftarrow \text{Reflect}(O, q_t, P_t, a_t, U_t)$ ;

        $\text{WriteToMemory}(O, r)$

    **end**

    Store experience $(q_t, \mathcal{G}_t, U_t, C_t)$ in $\mathcal{H}$;

**end**

---

# E    META-CONTROLLER IMPLEMENTATION

The meta-controller is trained using the REINFORCE algorithm, with a standard batch policy loss:

$$\mathcal{L}_{\text{batch}}(\theta) = -\frac{1}{N} \sum_{t=1}^{N} \log \pi_\theta(a_t|s_t) R_t \tag{6}$$

The reward $R_t$ is designed to balance performance, cost, and contextual appropriateness. It is a function of the final task score $s_t$, the total cost $C_t$, and the cosine similarity between the query and the current architecture, $\text{sim}(q_t, \mathcal{G}_{t-1})$.

The core of our reward function is a weighting mechanism that modulates the score $s_t$ based on this similarity. The reward for a RETAIN action is boosted when the architecture is a good match for the query (high similarity), while the reward for a RESAMPLE action is boosted when there is a

mismatch (low similarity). This can be expressed conceptually as:

$$R_t = s_t \cdot w(a_t, \text{sim}(q_t, \mathcal{G}_{t-1})) - \lambda C_t \tag{7}$$

where the weighting function $w(\cdot, \cdot)$ increases the effective reward for correct decisions. For example, $w(\texttt{RETAIN}, \text{sim})$ is an increasing function of similarity. This formulation provides a dense and informative signal that guides the meta-controller to learn an efficient, context-aware policy.

## F  DATASET STATISTICS

For each of the benchmarks, we follow established methodologies for workflow automation (Hu et al. (2025), Zhang et al. (2025b), Zhang et al. (2025c)) and use a train-to-test ratio of $1 : 4$. We select 19 subdomains for MMLU, spanning formal mathematics, biology, chemistry, clinical medicine, business, and engineering. For SciCode, we use the standard subproblem setup without prior scientist annotations and report the subproblem pass rate.

Table 3: Dataset statistics.

| Domain | Dataset | Train Samples | Test Samples | Metric |
|---|---|---|---|---|
| Math Reasoning | MATH | 100 | 400 | Accuracy |
| Question Answering | MMLU | 100 | 400 | Accuracy |
| | GPQA | 89 | 359 | Accuracy |
| Code Generation | HumanEval | 33 | 131 | Pass@1 |
| | SciCode (subproblems) | 51 | 287 | Pass@1 |

## G  PROMPTS

### G.1  ARCHITECT'S PROMPT

We used the following prompt for the Architect. The decision to define the architecture representation with mathematical notation was deliberate. We observed through preliminary experiments that providing a formal syntax, as opposed to a natural language description, makes the instructions for concepts like parallelism and aggregation less ambiguous for the LLM. This leads to more consistent and structurally valid outputs from the Architect.

---

**Architect Prompt**

You are a multi-agent architect $f_{\mathbb{A}}$ mapping a query and context to an agent architecture: $f_{\mathbb{A}} : (q, C) \mapsto G$.

Goal: propose or adjust a layered operator architecture that is robust, performs well now, and is generalizable to future queries.

Follow these steps:

- Decompose the query, pick an initial architectural pattern, and justify briefly.

- Think of 2 other alternative architectural patterns, consider all 3 options, and select the best one.

- If creating specialists, provide concise identity and bullet directives (no steps or formulas - you are encouraged to use different reasoning patterns and strategies like adversarial prompting, quality-diversity prompting, step-back prompting, multi-choice elimination, etc.)

- Use the recent sliding window experiences as guidance for your decisions.

---

Architecture representation: $A = [L_1, \dots, L_K]$ where layer $L_i = [o_{i,1}, \dots, o_{i,m_i}]$ lists operators executed in parallel. Execution is layerwise: let $x_1 = x$ and for $i = 1, \dots, K$, compute a layer output $h_i(x_i)$. If $|L_{i-1}| > 1$, include an Ensemble aggregator $g_i$ to combine parallel outputs: $h_i(x_i) = g_i(\{ o(x_i) \mid o \in L_{i-1} \})$. If $|L_{i-1}| = 1$, then $h_i(x_i) = o(x_i)$ for the unique $o \in L_{i-1}$. The input to the next layer is $x_{i+1} = h_i(x_i)$.

Query: [...]

Context for your decision:

- Operator pool: [...]

- Current architecture: [...]

- Allowed actions: [...]

- Recent sliding window experiences: [...]

XML formatting guide: [...]

where the allowed actions are conditional on the specific phase the system is in. During the specialist discovery phase, the full operator space is used:

- `CREATE_OPERATOR`: Defines a new specialist operator $O_{\text{spec}}$ and adds it to $\mathbb{O}_{\text{spec}}^{(t)}$.

- `COMBINE_OPERATORS`: Merges two specialist operators into a single, more general specialist.

- `PRUNE_OPERATOR`: Removes a specialist operator from $\mathbb{O}_{\text{spec}}^{(t)}$.

- `FINALIZE_ARCHITECTURE`: Commits to a final graph $\mathcal{G}_t$ and terminates the reasoning loop.

We then restrict the allowed actions to only `FINALIZE_ARCHITECTURE` during the specialist cultivation and evaluation phases.

## G.2 SPECIALIST SYNTHESIS PROMPTS

### G.2.1 CREATION

**Creation: Identity Synthesis Prompt**

You will propose diversified identity variants for a new specialist, `operator_name`, which is based on `base_template`.

Specialist description: `[...]`

Each identity should be a detailed second-person identity including their professional role (i.e., 'a particle physicist', do not include names), their fields of expertise (deep + broad), and a non-domain-specific reasoning heuristic that distinguishes them from other specialists.

Examples of reasoning heuristics:
- Works backwards from contradictory answers to identify wrong assumptions or equations.
- Never assumes anything not explicitly stated; always returns to first principles when confused.
- Builds multiple competing hypotheses simultaneously and tests them against evidence.
- Visualizes problems as interconnected networks of constraints and relationships.

Output the identity text starting with 'You are...'

---

**Creation: Directive Synthesis Prompt**

You will propose diversified directive variants for a new specialist, `operator_name`, which is based on `base_template`.

Specialist description: `[...]`

Create a bulleted list of methodological principles and reasoning approaches that this new specialist will follow. Do not provide specific formulas, step-by-step procedures, formatting instructions, or direct solutions. Focus on **how** the specialist should think and approach problems, not what specific steps to take.

Include strategic reasoning approaches like self-criticism, assumption questioning, hypothesis building, pattern recognition, systematic analysis, etc. It is very important that the directives should guide analytical thinking without restricting the specialist's reasoning search space.

---

### G.2.2 CROSSOVER

---

**Crossover: Identity Synthesis Prompt**

You will propose diversified identity variants for a combined specialist that synthesizes the expertise of two parent specialists. The specialist is `operator_name`, which is based on `base_template`.

Specialist description: `[...]`
Parent 1's identity: `[...]`
Parent 2's identity: `[...]`

Each identity should be a detailed second-person identity including their professional role (i.e., 'a particle physicist'), their fields of expertise (deep + broad), and a non-domain-specific reasoning heuristic. The combined identity should integrate the best aspects of both parent specialists while creating a coherent, unified specialist persona.

Examples of reasoning heuristics:
- Works backwards from contradictory answers to identify wrong assumptions or equations.
- Never assumes anything not explicitly stated; always returns to first principles when confused.
- Builds multiple competing hypotheses simultaneously and tests them against evidence.
- Visualizes problems as interconnected networks of constraints and relationships.

Output the identity text starting with 'You are...'

---

**Crossover: Directive Synthesis Prompt**

You will propose diversified directive variants for a combined specialist that synthesizes the expertise of two parent specialists. The specialist is `operator_name`, which is based on `base_template`.

Specialist description: `[...]`
Parent 1's identity: `[...]`
Parent 2's identity: `[...]`

Create a bulleted list of methodological principles and reasoning approaches that this new specialist will follow. Do not provide specific formulas, step-by-step procedures, formatting instructions, or direct solutions. Focus on how the specialist should think and approach problems, not what specific steps to take. The combined directives should integrate the best aspects of both parent specialists' directives, including any existing reasoning approaches.

Include strategic reasoning approaches like self-criticism, assumption questioning, hypothesis building, pattern recognition, systematic analysis, etc. It is very important that the directives should guide analytical thinking without restricting the specialist's reasoning search space.

---

## G.3 JUDGE PROMPTS

---

**Judge Prompt: Evaluating Identities**

You are judging specialist identities for: `operator_name`, which is based on `base_template`.

Specialist description: `[...]`
Identity candidates: `[...]`

Pick the best identity based on the following criteria:
1. Non-domain-specific reasoning heuristics for a rich reasoning 'gene' pool (quality-diversity, step-back analysis, assumption-challenging, etc.)
2. Avoids making assumptions not explicitly stated in the problem
3. The resulting specialist is a T-shaped specialist. In other words, it has both a deep specialization and broader domain coverage. Avoid hyperspecific specialists that are too narrow in their domain coverage.
4. Combines domain expertise with general problem-solving approaches

---

**Judge Prompt: Evaluating Directives**

You are judging specialist directives for: `operator_name`, which is based on `base_template`.

Specialist description: `[...]`
Directive candidates: `[...]`

Pick the best directives based on the following criteria:
1. Focus on how to think, not what specific steps to take. Mimic domain-specific human experts to guide analytical thinking without constraining solution paths
2. Prefer methodological principles over rigid instructions. Avoid specific formulas, procedures, or direct solutions
3. Has a specific methodology for handling contradictions or confusion

---

