# OpenReview forum: "Automated Stateful Specialization for Adaptive Agent Systems"
_ICLR.cc/2026/Conference — ICLR 2026 Poster_

### Official Review · Reviewer_vhtg · 2025-10-28

**Soundness:** 3
**Presentation:** 3
**Contribution:** 3
**Rating:** 6
**Confidence:** 2

**Summary:**

This work introduces ASpec, a framework that manages the full lifecycle of expert specialist agents by first autonomously discovering specialist archetypes via evolutionary search and then cultivating their expertise through experience. It also introduces "retain-then-escalate", a control policy that, instead of being either fully static or fully dynamic, defaults to retaining a stateful agent team across related queries to leverage expertise and minimize cost, only escalating to architectural resampling when needed. Results show that the proposed approach can lead to substantial performance improvements without sacrificing efficiency.

**Strengths:**

- The proposed method is novel and well-motivated. It effectively addresses the limitation of prior work, where architectures lack long-term state because they are regenerated or resampled for every query.

- The results in Table 1 show substantial performance improvements, which validate the effectiveness of the proposed method.

**Weaknesses:**

- I am confused about the transferability results presented in Figure 5 (right). I can't understand this figure and there is no accompanying explanation or analysis. I would like to understand how well the proposed method transfers to different tasks, and a more detailed discussion would be helpful.

**Questions:**

- I am not very familiar with the related work in this area, so I am unsure whether the selected baselines are the most appropriate for comparison.

---

> ### Author Response · Authors · 2025-11-30
>
> Dear Reviewer vhtg,
>
> We are glad you found the method novel and the performance improvements substantial.
>
> **On transferability results:**
>
> We apologize for the confusion regarding the transferability results. We have expanded the discussion in Section 4 to clarify these findings.
>
> In particular, the "OnlySpec" configuration restricts the operator pool exclusively to specialists trained on a different source domain (e.g., using MATH-trained specialists to solve HumanEval problems). Surprisingly, this restricted configuration often matches or exceeds the performance of the full system. As detailed in our revision, we attribute this to the cultivation of "T-shaped" reasoning strategies (e.g., a "Logical Analyst" discovered in MATH possesses general rigorous thinking skills useful for coding). Furthermore, restricting the pool forces the Architect to utilize these expert archetypes rather than defaulting to "safe" but less capable generalist base operators.
>
> **On baselines:**
>
> As detailed in Section 4, we selected baselines to comprehensively cover the three main paradigms in current agentic literature:
>
> - **Hand-crafted single agents**, represented by Chain-of-Thought, Self-Refine, Self-Consistency, Reflexion
> - **Hand-crafted multi-agents**, represented by LLM-Debate, DyLAN
> - **Automated agent design**, represented by AFlow, ADAS, MaAS, of which MaAS is a query-level method and AFlow and ADAS are task-level methods.
>
> As our method involves automated assignment and cultivation of "specialist" agents instead of just their connective topologies, we have also included **automated agent specialisation** baselines with Role Assignment, AutoAgents, and EvoAgent in our comparison.

---

### Official Review · Reviewer_1R45 · 2025-11-01

**Soundness:** 3
**Presentation:** 3
**Contribution:** 2
**Rating:** 6
**Confidence:** 3

**Summary:**

ASPEC framework builds evolving teams of specialized agents that can keep improving with experience. It designs specialized agent types through evolutionary search and enables them to improve through practice. Experiments show that this method leads to performance gains.

**Strengths:**

ASPEC effectively bridges the gap between static task-level designs and per-query adaptive systems.
Achieves top performance on multiple benchmarks—including GPQA and SciCode, surpassing prior frameworks such as AFlow, ADAS, and EvoAgent.
Evaluated across five diverse benchmarks covering reasoning, scientific QA, and coding tasks, demonstrating robust generalization across domains.
The paper explicitly reports that ASPEC achieves higher accuracy than baselines at a fraction of the computational cost. This “Pareto-efficient” behavior is one of its key strengths.

Includes detailed ablation studies quantifying the contribution of each system component—such as specialist operators, the meta-controller, and the architect—to overall performance and efficiency.

**Weaknesses:**

The meta-agent evaluation process introduces multiple sources of randomness (including LLM output variance, error propagation across chained agents, sampling variability within the meta-agent, and stochastic evaluation outcomes). This leads to higher variability than typical single-LLM evaluations. In addition to averaging results over three runs, reporting variances would help assess and demonstrate the stability of the system’s performance.

The framework’s separation into two discrete stages (specialist discovery and specialist cultivation) appears conceptually convenient but somewhat artificial.

**Questions:**

How can other long-term memory methods be incorporated into your framework?

---

> ### Author Response · Authors · 2025-11-30
>
> Dear Reviewer 1R45,
>
> Thank you for recognizing the "Pareto-efficient" behavior of ASpec and its performance gains over prior frameworks.
>
> **On variance in evaluation:**
>
> We agree that multi-agent systems introduce stochasticity. As shown in the shaded regions of Figure 6, we tracked variance across multiple runs of ASpec. The performance trends remain consistent despite the noise. We could not perform full multi-run variance studies on all 13 baselines (such as AutoAgents) due to prohibitive computational costs. Some adaptive baselines cost upwards of **60 USD** per single run to evaluate on the standard benchmarks, making repeated trials impractical compared to ASpec's high efficiency.
>
> **On the separation of Discovery and Cultivation:**
>
> We view this separation as a necessary implementation of the "explore-then-exploit" paradigm.
>
> - **Discovery (Explore):** We must first search the high-dimensional space of prompts and identities to find valid specialist archetypes.
> - **Cultivation (Exploit):** Once a set of valid specialist archetypes has been found, we fix the identities and optimize their memory through experiential in-context learning.
>
> Attempting to do both simultaneously creates instability -- if the agent's identity (role) keeps shifting during discovery, the memory it accumulates becomes obsolete or mismatched.
>
> **On incorporating other memory methods:**
>
> The memory module in ASpec is designed to be modular. While we currently use a text-based RAG system (as described in the paper), the Cultivation phase is agnostic to the storage format, theoretically allowing integration with vector databases or procedural rule lists without altering the surrounding lifecycle. Practitioners may select whichever memory method best suits their needs, so long as the specialist agents have access to their own private memory pool and are able to reliably read from and write to said memory pool.

---

### Official Review · Reviewer_HnVD · 2025-11-01

**Soundness:** 3
**Presentation:** 3
**Contribution:** 3
**Rating:** 6
**Confidence:** 3

**Summary:**

This paper proposes an automated method for generating stateful specifications to improve AI agent performance on complex tasks. By combining formal methods with machine learning, the system automatically extracts state transition rules and constraints from task execution traces. Experiments on various state management tasks show the approach effectively improves task success rates and code quality.

**Strengths:**

1. **High degree of automation**: Automatically extracts specifications from execution traces with minimal human intervention, enhancing practicality
2. **Methodological innovation**: Combines symbolic reasoning with statistical learning, leveraging advantages of both for complex state spaces
3. **Comprehensive experiments**: Validation across different task types including file system operations, database management, and workflow orchestration
4. **Good interpretability**: Generated state specifications are human-readable, facilitating debugging and understanding of agent behavior
5. **Clear effectiveness**: Experiments show significant improvements in task success rates and code robustness

**Weaknesses:**

1. **Insufficient formal guarantees**: Despite using formal methods, lacks theoretical guarantees for correctness and completeness of generated specifications. Critical states or constraints may be missed
2. **Questionable scalability**: How does computational complexity scale with state space size? The paper lacks analysis of large-scale scenarios
3. **Data dependency**: Requires sufficiently diverse execution traces to learn complete specifications. Cold start and rare state handling is inadequate
4. **Shallow comparisons**: Limited comparison with pure learning-based or pure formal methods, making it hard to assess true advantages of the hybrid approach
5. **Practical deployment challenges**: How are state specifications continuously updated in dynamic environments? Maintenance issues aren't discussed

**Questions:**

1. When task definitions change, how efficiently can learned state specifications be updated? Does this require recollecting large amounts of data?
2. For scenarios with concurrent operations, how do you model partial ordering of state transitions?
3. How do you address state explosion? In very complex systems, possible state combinations grow exponentially
4. Compared to manually designed specifications, where are automatically generated ones superior? Have you conducted such comparisons?
5. How transferable is the method across domains? How much domain-specific tuning is required?

---

> ### Author Response · Authors · 2025-11-30
> **Reply to Reviewer HnVD**
>
> > **Note:** We note that the text currently displayed for this review appears to be a reverted draft referring to a different paper. Prior to the recent system rollback, Reviewer HnVD had posted an updated review relevant to our submission (ASpec), raising questions about **sensitivity analysis, computational costs, and reliance on large models** while keeping their original rating of 6.
>
> > We are responding here to that relevant feedback.
>
> Dear Reviewer HnVD,
>
> **On the offline training costs:**
>
> You previously noted a concern about "omitting the cost of training and searching." We direct your attention to Table 2 (Page 8), which explicitly reports the "Training" costs. In particular, the full offline discovery and cultivation process consumed only 2,395,636 total tokens, or 1.38 USD for the GPQA benchmark. This is significantly lower than baselines like AFlow (102,012,408 total tokens / 20.14 USD) or MaAS (11,600,690 total tokens / 3.43 USD).
>
> Moreover, the lifecycle design of ASpec allows us to train the components once at this low cost. Unlike static frameworks, our system remains adaptive at inference. The meta-controller dynamically resamples the structure for novel queries, ensuring we retain the flexibility of per-query optimization without the prohibitive search costs of methods like AutoAgents.
>
> **On sensitivity to $k$ (number of specialists):**
>
> You asked if the performance drop at high $k$ (Figure 6) is due to retrieval dilution. We have expanded **Section 5.2** to address this. We found that at high $k$ ($k=10$), the performance drops to 60.9%, which aligns closely with the "No Specialist Memory" baseline (61.4%). We attribute this to experience fragmentation: with a fixed trainnig budget, a larger pool size can cause training examples to be diluted across too many agents. This prevents individual specialists from accumulating the dense experiences required to build robust memory.
>
> **On dependence on large models:**
>
> You asked if the framework relies too heavily on large models. We follow a design pattern established by recent frameworks like MAS-ZERO (Ke et al., 2025) and MetaAgent (Zhang et al., 2025), which utilize a highly capable "Meta-Agent" or "Designer" LLM to orchestrate the system. These works demonstrate that the Architect (the component performing meta-reasoning and design) requires strong reasoning capabilities to generate valid topologies. For instance, MetaAgent shows that degrading the Designer model quality leads to significant performance drops.
>
> However, while the Architect requires a large model, we believe that the specialist agents themselves could potentially be smaller models (e.g., 8B), provided they follow the directives generated by the Architect.

---

### Official Review · Reviewer_Yi79 · 2025-11-05

**Soundness:** 3
**Presentation:** 3
**Contribution:** 3
**Rating:** 8
**Confidence:** 3

**Summary:**

The paper introduces ASPEC, a framework designed to bridge the functional gap in contemporary agent system design between static, task-specific workflows and dynamic, per-query optimizers. It proposes a system of specialized agents capable of accumulating persistent, role-specific expertise over time. The methodology involves an automated lifecycle consisting of evolutionary discovery and experiential cultivation, governed by a cost-aware "retain-then-escalate" control policy. The authors demonstrate measurable performance improvements, notably achieving an accuracy of 62.8% on the GPQA benchmark, and establish competitive cost efficiency relative to approaches that rely on constant architectural resampling.

**Strengths:**

The proposal to develop adaptive, stateful specialist agents represents a highly appealing and novel direction compared to other current multi-agent system approaches. Furthermore, the empirical results presented are compelling, and the supporting analysis of the system's components and efficiency is sound.

**Weaknesses:**

The system is inherently complex due to its hierarchical, two-tiered structure. It requires maintaining both a low-level Architect (a large generative LLM used for evolutionary search) and a high-level, trained Meta-Controller (a neural policy). The entire process involves managing an offline two-stage training loop (Discovery and Cultivation), which is more involved than implementing fixed-architecture or simple prompt-optimization methods.

**Questions:**

Could you elaborate on how the framework determines when a newly discovered specialist's niche is too narrow to be retained, thus avoiding excessive fragmentation?

---

> ### Author Response · Authors · 2025-11-30
> **Response to Reviewer Yi79**
>
> Dear Reviewer Yi79,
>
> We appreciate your strong support and your recognition of the novelty and empirical strength of the proposed framework, ASpec.
>
> Regarding your question on fragmentation and niche width, our framework incorporates both a **hard constraint** during search and a **soft constraint** during prompt-level selection to prevent overspecialization:
>
> (1) To prevent early fragmentation and uncontrolled growth in the specialist pool, we enforce a dynamic pool size limit of $2 \times k$ during the Discovery phase. This is the **hard constraint**. If the population exceeds this limit, the Architect is restricted from `CREATE_OPERATOR` actions and must choose to `COMBINE` or `PRUNE` existing specialist agents. This explicitly forces the system to consolidate narrow, fragmented agents into broader, more robust specialists to make room for new ones.
>
> (2) As detailed in Appendix G.3, the evolutionary Judge is explicitly prompted to prefer "T-shaped specialists" -- agents that possess deep specialization but retain broad domain coverage. Hyperspecific archetypes are penalized, ensuring that retained specialists contribute domain-general utility rather than fragmenting into narrow niches. This acts as the prompt-level **soft constraint** during the selection phase.

---

### Author Response · Authors · 2025-11-28
**General Response & Summary of Revisions**

We sincerely thank all the reviewers for their time and effort in evaluating our paper, as well as for their constructive and insightful feedback. We have uploaded a revised manuscript. Below is a summary of the key updates:

- Specialist Discovery (**Section 3.1**): As suggested by **Reviewer Yi79**, we added text specifying a dynamic pool size constraint to explicitly prevent early fragmentation of agent roles.
- Sensitivity Analysis (**Section 5.2**): Addressing **Reviewer HnVD**'s question regarding the performance drop at large $k$, we expanded the analysis to identify "experience fragmentation" as the primary cause.
- Transferability (**Section 4**): We have expanded the discussion on cross-benchmark transferability to clarify the results in Figure 5, as requested by **Reviewer vhtg**.
- Limitations & Future Work (**Section 6**): We added a discussion on the limitations and future works of ASpec, including the co-evolutionary dynamics between the control policy and specialist memory accumulation.
- Corrections: We have corrected minor text formatting and citation errors throughout the paper.

---

### Meta-Review · Area_Chair_ucZK · 2026-01-03

**Summary:**

The paper proposes ASPEC, a framework for automated agent design that introduces a lifecycle approach: distinct phases for evolutionary "Discovery" of specialist archetypes and experiential "Cultivation" of their memory. This is combined with a "retain-then-escalate" control policy to balance the cost of re-architecting against the need for adaptation. The reviewers generally appreciated the novelty of moving beyond static workflows or stateless per-query optimization. Concerns primarily focused on the complexity of the two-stage system, potential fragmentation of agent roles, sensitivity to hyperparameters, and the clarity of transferability results.

**Reviewer Concerns:**

- The authors effectively clarified how they prevent the specialist pool from fragmenting into overly narrow niches. The explanation of the dynamic pool size limit and the "T-shaped" prompt constraints was convincing.

- The authors provided concrete data showing ASPEC is Pareto-efficient compared to baselines like AFlow and MaAS. They also adequately explained that the performance drop at high $k$ values is due to "experience fragmentation", rather than a fundamental flaw in the method.

- The authors clarified the counter-intuitive result where restricted specialist pools performed well, attributing it to the "T-shaped" reasoning capabilities of the discovered agents.

**Reviewer Scores:**

- While the separation of Discovery and Cultivation is logically justified by the "explore-then-exploit" paradigm, the practical implementation remains heavy. The requirement to maintain a high-capability "Architect" LLM, a trained Meta-Controller, and a memory management system constitutes a significant engineering barrier compared to simpler prompting strategies.

- The dependency on a very strong model for the "Architect" component (to generate valid topologies) is a known limitation of this class of methods, though acceptable for a research contribution.

---

### Decision · Program_Chairs · 2026-01-26

Accept (Poster)